# Alternative ecological strategies lead to avian brain size bimodality in variable habitats

Trevor S. Fristoe [1,2] & Carlos A. Botero [2]

The ecological contexts that promote larger brains have received considerable attention, but those that result in smaller-than-expected brains have been largely overlooked. Here, we use a global sample of 2062 species to provide evidence that metabolic and life history tradeoffs govern the evolution of brain size in birds and play an important role in defining the ecological strategies capable of persisting in Earth's most thermally variable and unpredictable habitats. While some birds cope with extreme winter conditions by investing in large brains (e.g., greater capacity for planning, innovation, and behavioral flexibility), others have small brains and invest instead in traits that allow them to withstand or recover from potentially deadly events. Specifically, these species are restricted to large body sizes, diets consisting of difficult-to-digest but readily available foods, and high reproductive output. Overall, our findings highlight the importance of considering strategic tradeoffs when investigating potential drivers of brain size evolution.

[1] Department of Biology, University of Konstanz, Universitätsstraße 10, 78464 Konstanz, Germany. [2] Department of Biology, Washington University in St. Louis, Campus Box 1137, One Brookings Drive, St. Louis, MO 63130-4899, USA. Correspondence and requests for materials should be addressed to T.S.F. (email: trevor.fristoe@uni-konstanz.de)

Coping with environmental variability can be challenging[1], particularly if climatic conditions oscillate between opposing extremes (e.g., hot summers and cold winters)[2]; resources vary in type, abundance, or accessibility[3–5]; and habitat structure changes dramatically within a lifetime[6]. These ecological challenges are further amplified when environmental change is unpredictable because the ability to anticipate it through phenological shifts in physiology, morphology, or behavior is compromised[7]. In these situations, individuals may partially buffer the negative effects of novel, extreme, or unexpected conditions through the rapid deployment of flexible or innovative behavioral responses. Accordingly, the cognitive buffer hypothesis posits that variable and unpredictable environments should favor enhanced encephalization (i.e., larger brains relative to body size) despite the high energetic and developmental costs associated with investment in neural tissue[8].

Recent comparative studies on birds have lent support to these ideas by showing that highly encephalized lineages are overrepresented in cold, seasonal, and unpredictable high-latitude habitats[9,10] and are better able to maintain more stable populations when conditions vary[11] as compared to their small-brained counterparts. Nevertheless, not all birds that reside in these habitats year-round have relatively large brains. For example, the grouse (subfamily *Tetraoninae*) occur widely throughout highly seasonal and thermally unpredictable North temperate and arctic habitats[12] but possess some of the smallest relative brain sizes known among birds[13]. Perhaps more surprising is the recent suggestion that, among Galliformes (the larger taxonomic unit to which the grouse belong), there may be a trend toward smaller-than-expected brain sizes in increasingly variable environments[9]. These conspicuous exceptions to the patterns predicted by the cognitive buffer hypothesis have received little attention in the scientific literature, reflecting a possible anthropocentric bias toward investigating the evolutionary forces directly relevant to our own encephalization[8,14]. However, a better understanding of the evolution of brain size and cognition is likely to result from increased clarity on the ecological contexts that promote both relative brain enhancements and reductions.

Here we re-examine the relationship between environmental conditions and relative brain size in birds and evaluate the extent to which the presence of small-brained species in variable and unpredictable environments can be regarded as statistical noise. We begin by showing that, in contrast to birds living in more climatically stable regions of the world, resident birds in thermally variable habitats exhibit an overrepresentation of both very small and very large brains and a nearly complete absence of intermediately sized brains. We then provide initial evidence supporting the idea that this striking morphological pattern can be linked to the presence of alternative eco-morphological strategies that cope with the inherent challenges of extreme winter conditions by emphasizing different aspects of ecological and life history trade-offs imposed by the high metabolic demands of brain tissue.

## Results

**The biogeography of brain size**. To gain a better understanding of how the distribution of relative brain size varies geographically, we collated a comprehensive global sample of publicly available data on non-migratory bird species[11] ($N = 1280$). We first grouped species by occurrence within six geographic regions delineated by climactic conditions (Fig. 1). Because climatic variables are known to be highly correlated[15], we defined these six environmental regions using a composite metric defined as the first component (PC1) of a geographic principal component analysis based on a 100+ year time series dataset of monthly variation in temperature and precipitation (Supplementary Table 1, see "Methods")[16]. PC1 captured a gradient of variation in the mean, seasonality, and predictability of temperature, with high values indicating warmer, more stable, and typically tropical habitats and low ones indicating colder, more variable, and typically high-latitude ones. A second component in this analysis (PC2) included variation in the mean, predictability, and relative annual variability of precipitation, capturing a different aspect of environmental variability. We do not discuss PC2 further in the main text of this article because it was not found to be meaningfully associated with geographic variation in brain size distribution (Supplementary Fig. 1).

Our data indicate that the relative brain size of resident birds is approximately normally distributed throughout the world except in high-latitude regions characterized by harsh winters and highly variable and unpredictable temperatures. In those regions, large and small brain sizes are significantly overrepresented, and intermediate brain sizes are conspicuously absent (Fig. 1). To investigate the extent to which these apparent distributional differences could have arisen by chance, we used randomization tests[17] based on null distributions of relative brain size, derived for each environmental region by drawing 10,000 samples of $n$ species (the number of species in the region of interest) without replacement from the global species pool (see "Methods" for details). Our analyses confirm that obtaining the seemingly bimodal pattern observed at high latitudes by chance is highly unlikely, even considering the relatively low species richness of these regions (Fig. 1). Furthermore, the brain size distributions observed in those regions is unlikely to have resulted from incomplete taxonomic sampling because our sample does not disproportionally omit resident clades with intermediate brain sizes (Supplementary Fig. 2). Thus the currently available data support the notion that there are two viable brain size strategies for dealing with these thermally variable environments and strongly suggest that the presence of small-brained species in these challenging habitats represents a pattern of true ecological significance rather than statistical noise.

**Trade-offs and the maintenance of alternative strategies**. In contrast to residents, migratory species that breed at high latitudes exhibit a fairly Gaussian brain size distribution that does not significantly deviate from null expectations (Fig. 2; total sample of $n = 623$ species). This finding suggests that the alternative brain size strategies found among resident birds in these regions are likely to be a consequence of the challenges experienced during the resource-poor and thermally extreme winter months[10,18]. Such a possibility is consistent with the observation that well-known cases of alternative life history strategies have evolved from trade-offs imposed by limiting resources[19]. Brain tissue is both metabolically and developmentally costly[20,21], so it is likely that investments in larger brains will limit the deployment or use of alternative solutions to the challenges of harsh winters[22]. Thus small-brained species in variable habitats may allocate a greater proportion of their limited budget to other costly morphological and life history traits that facilitate persistence in these difficult environments. In the following two sections, we evaluate support for possible trade-offs between brain size and additional traits, specifically diet and reproductive output, and quantify the extent to which these trade-offs are further constrained by the extreme winter conditions experienced at high latitudes.

**Resource reliability: trade-offs in diet**. One possible way to minimize the impact of environmental fluctuations and winter scarcity is to specialize on resources that show little variation in

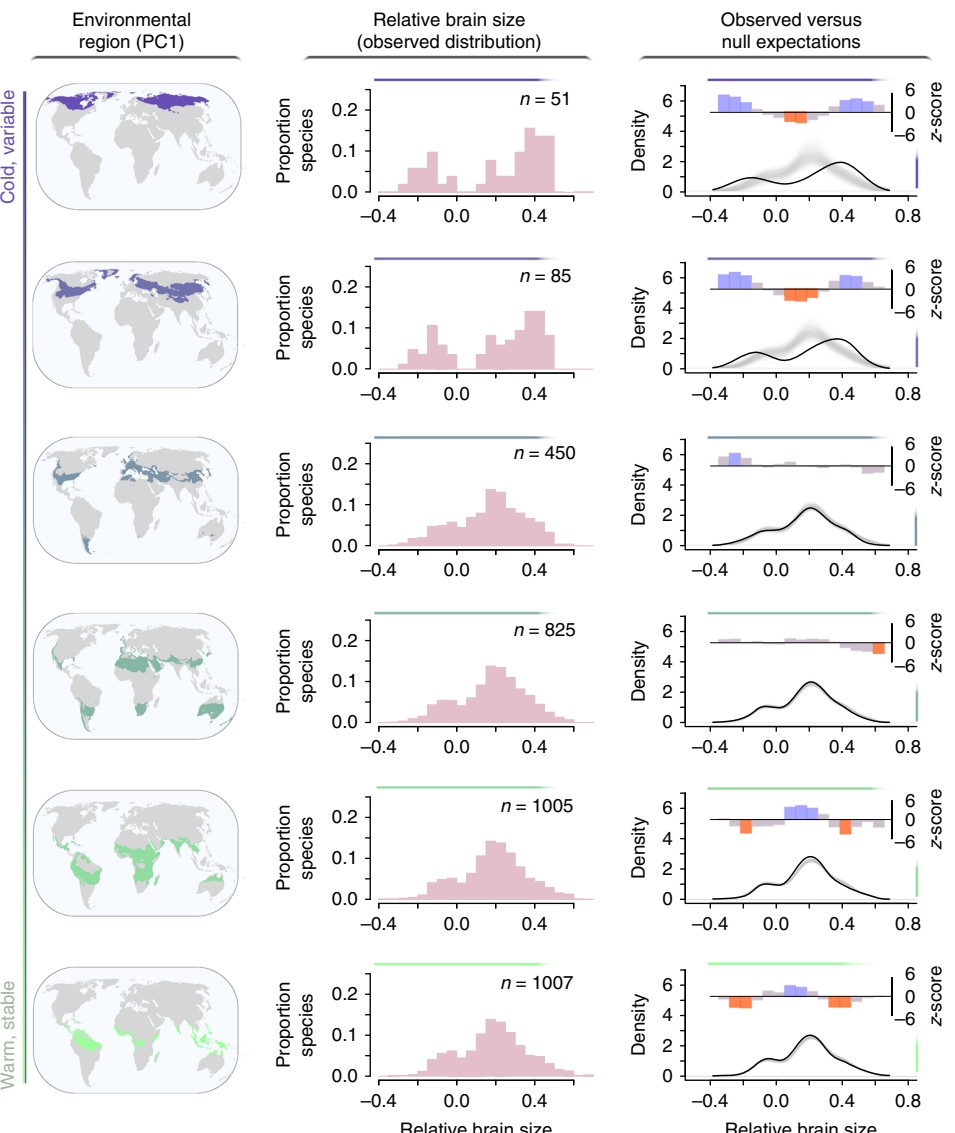

**Fig. 1** The global distribution of brain size for resident birds in relation to environmental PC1. The observed distributions of relative brain size for 1280 species (middle column) within environmental bins (left column) are compared to null expectations (right column). Colored areas of maps depict the regions with conditions falling within a given window of composite environmental variable PC1 (see "Results" section of the text); regions of cold, seasonal, and unpredictable temperatures are in dark purple (low PC1 scores) and warm, stable regions in green (high PC1 scores)[16]. Each observed frequency distribution of relative brain size includes all species with breeding distributions that overlap with the given environmental region. Kernel density estimates of relative brain size for the observed sample of $n$ species that occur within each environmental region (solid black lines in right column) are compared to 10,000 null density estimates, each derived from $n$ randomly sampled species (1000 examples plotted as light gray lines). At 15 evenly spaced points across the full range of relative brain size values, empirically derived density estimates are compared to density estimates from the null samples; the height of bars in the third column depict $z$-scores from these comparisons. For relative brain sizes that were observed at a significantly higher than expected frequency within a given environmental region, bars are depicted in blue, those observed significantly less frequently are depicted in red, and those that do not differ significantly from null expectations are depicted in gray. Source data are provided as a Source Data file

availability as a function of climate[23,24]. Buds, twigs, and conifer needles tend to be abundant year-round in many high-latitude habitats and are typically easily accessible even when snow is plentiful. However, these readily available food items are potentially incompatible with the high metabolic demands of large brains because they are fibrous and require a large, energetically costly gut to digest[25,26]. Indeed, a trade-off between diet and brain size is evidenced in our global sample of resident birds. For example, we found that relative brain size appears to place an upper- but not a lower-bound constraint on the percentage of vegetative plant material in a species' diet (Fig. 3a; $n = 1304$; upper quantile regression line: $\tau = 0.9$, $\beta \pm SE = -23.78 \pm 9.35$,

$p = 0.01$; lower quantile: $\tau = 0.1$, $\beta \pm SE = 0 \pm 0$, $p = NA$). Specifically, while small-brained species can include any amount of vegetative plant material in their diet (even to the point of being able to subsist almost exclusively on these food items), large-brained ones cannot.

To investigate whether the ability to rely on low-quality diets is important for the persistence of small-brained species in thermally variable habitats, we began by dividing our sample into species that consume low versus high proportions of vegetative plant material and have small versus large relative brain sizes, setting cutoffs for these two traits at their corresponding mean values in our entire global sample. Across

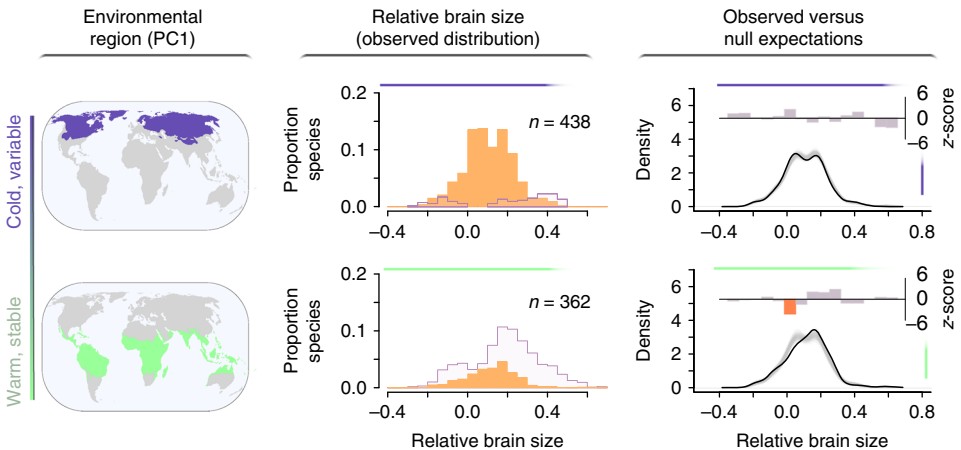

**Fig. 2** The global distribution of brain size for migratory birds in relation to environmental PC1 (i.e., a metric of temperature variability and unpredictability). The observed distributions of relative brain size for 623 species (middle column, in orange) within two environmental regions (left column) are compared to null expectations (right column). The map regions colored in deep purple are those with the lowest values of environmental PC1 and are characterized by cold and variable temperatures; the regions in green, with high values of environmental PC1, are warm and stable[16]. Each observed distribution of relative brain size includes all migratory species with breeding distributions that overlap with the corresponding environmental region; the distribution of relative brain size for resident species that live year-round in each environmental region is included in outline for reference. Kernel density estimates of relative brain size for the observed sample of *n* migratory species that occur within each environmental region (solid black lines in right column) are compared to 10,000 null density estimates, each derived from *n* randomly sampled migratory species (1000 examples plotted as light gray lines). At 15 evenly spaced points across the full range of relative brain size values, empirically derived density estimates are compared to density estimates from the null samples; the height of bars in the third column depict *z*-scores from these comparisons. For relative brain sizes that were observed at a significantly higher than expected frequency within a given environmental region, bars are depicted in blue, those observed significantly less frequently are depicted in red, and those that do not differ significantly from null expectations are depicted in gray. Source data are provided as a Source Data file

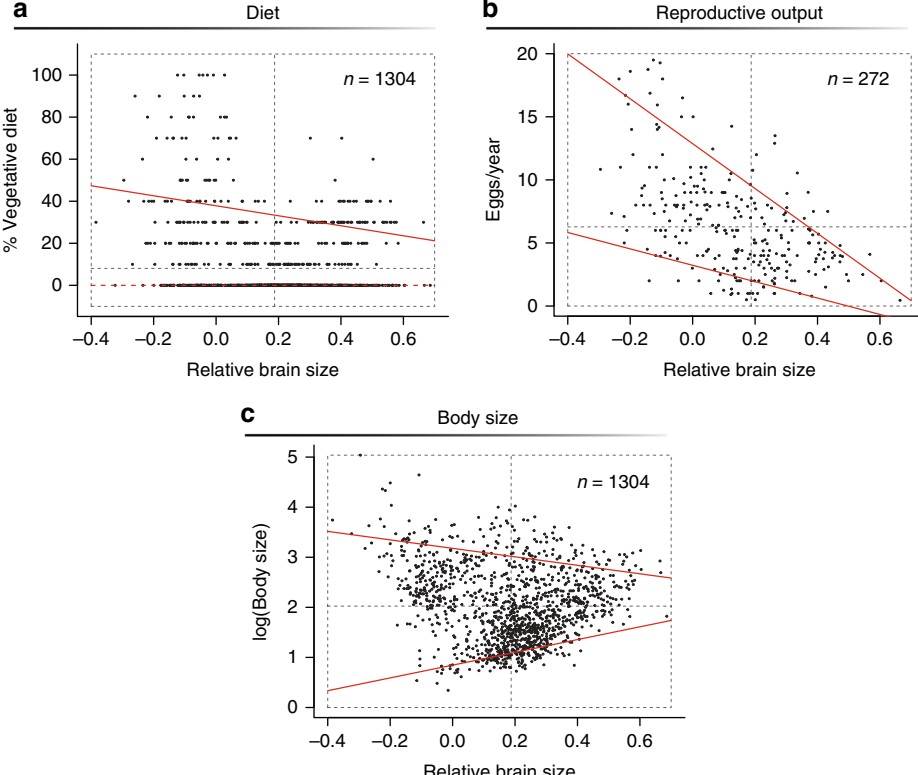

**Fig. 3** Constraints imposed by relative brain size on additional traits. For the global sample of resident species, quantile regression with $\tau = 0.9$ and $\tau = 0.1$ was used to test whether upper or lower values of diet (**a**), reproductive output (**b**), and body size (**c**) change with relative brain size; slopes that differ significantly from zero are indicated with a solid red line, those that do not with a dashed red line. For reference, dotted gray lines show the mean value for each trait and define the quadrants used in the analyses depicted in Fig. 4. Source data are provided as a Source Data file

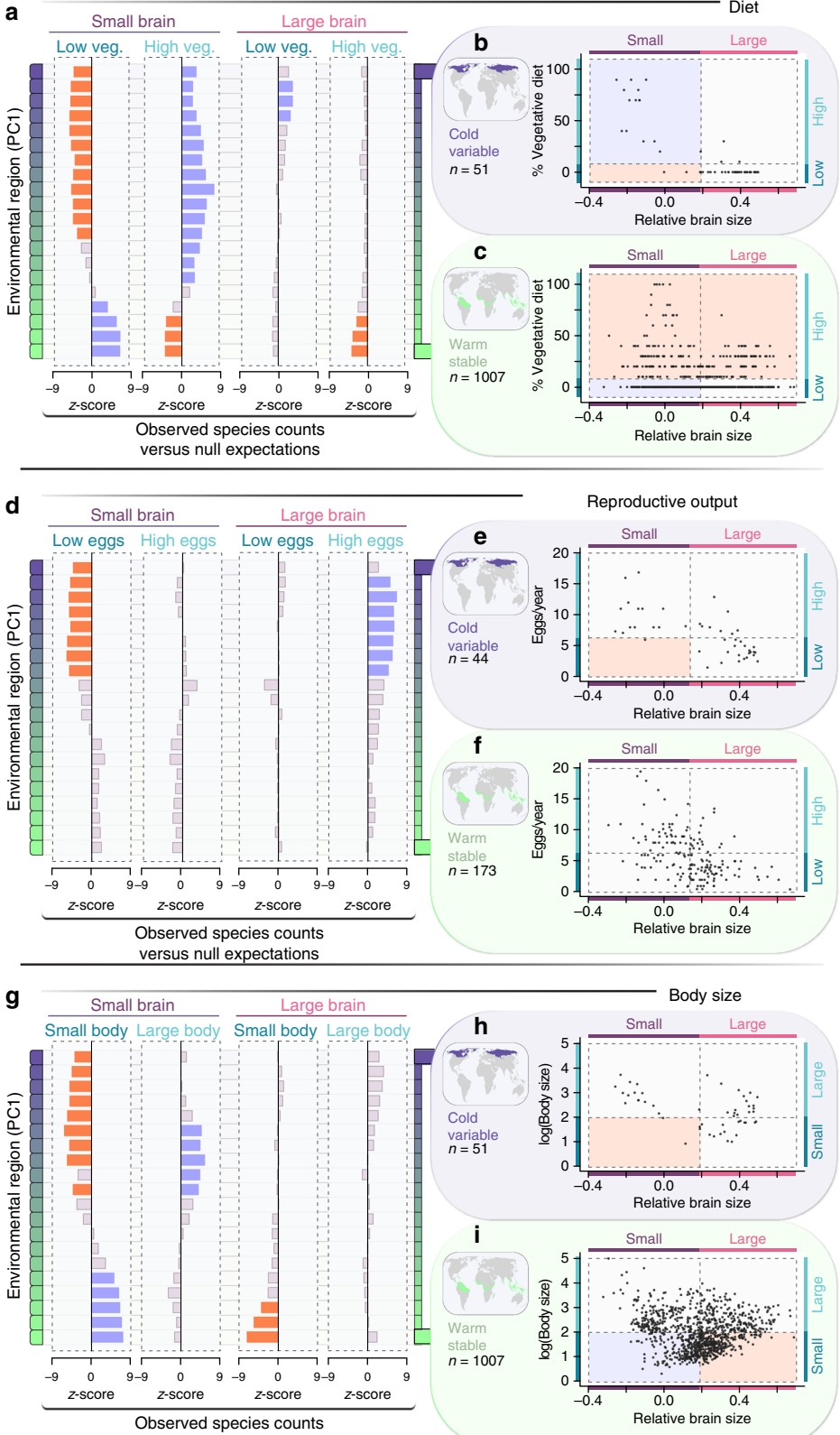

a series of environmental regions spanning the range of PC1, we identified all species that occur within the region, counted the number of species possessing each trait combination, and compared these numbers to null expectations derived through randomization (Fig. 4; see "Methods"). As before, the distribution of expected species counts for each trait combination was derived

by drawing 10,000 samples of *n* species from the global species pool, where *n* was the total number of species that occur in the environmental region of interest. Among small-brained birds, we found that, while tropical species exhibit higher-than-expected diet quality (Fig. 4a, c), species in cold, thermally variable climates include almost exclusively those that are able to subsist on

**Fig. 4** Environmentally imposed constraints on trait combinations. Changes in the frequency of different trait combinations are shown for a sample of 1280 resident species along an environmental gradient. Each row in **a**, **d**, **g** represents an environmental region along composite variable PC1 (see "Results" section of the text); regions with warm, stable temperatures are in green (high PC1 scores) and those with cold, variable temperatures are in deep purple (low PC1 scores)[16]. Each column represents a different trait combination: either small or large relative brain size, paired with a low or high vegetative diet (**a**), low or high reproductive output (**d**), or small or large body size (**g**); global mean values were used to discretize each trait). For all resident species whose geographic distributions overlap a given environmental region, we compared the numbers that possess each trait combination to null expectations (see "Methods" section of the text). The size of bars in **a**, **d**, **g** depict z-scores from these comparisons, with trait combinations occurring at significantly higher frequency than expected within a given environmental region in blue, those occurring significantly less frequently in red, and those that do not differ from null expectations in gray. Panels on the right (diet: **b**, **c**; reproductive output: **e**, **f**; body size: **h**, **i**) show trait values for species occurring within the two most extreme environmental regions (colored areas on maps). Dashed lines indicate the global mean value for each trait and divide trait space into quadrants; each quadrant corresponds to one trait combination (column) in left panels. The background color for each quadrant shows the result of statistical analyses depicted in the corresponding row and column of **a**, **d**, or **g**. Source data are provided as a Source Data file

vegetative plant material (Fig. 4a, b). Large-brained species, on the other hand, were generally found to be those that include only small proportions of vegetative plant material in their diets, particularly within the tropics, where the number of species with low-quality diets is significantly lower than expected by chance (Fig. 4a–c).

**Offsetting mortality: trade-offs in reproductive output.** Factors other than survival can also influence persistence in ecologically challenging environments. For example, populations of species that exhibit high mortality rates during periods of climatic stress may nevertheless be resilient to environmental oscillations if they have a strong ability to recover once more favorable conditions return[27,28]. Brain size has been implicated in a number of life history trade-offs that influence reproductive output and population growth[29]. Specifically, large-brained species typically require long and costly periods of parental care[30,31], limiting their ability to raise a large number of young. Accordingly, a negative upper-bound constraint in the relationship between reproductive output and brain size observed in our global sample of resident birds (Fig. 3b; $n = 272$; upper quantile regression line: $\tau = 0.9$, $\beta \pm SE = -17.78 \pm 1.88$, $p \ll 0.001$) is indicative of a strong trade-off. In contrast, a shallower slope for the lower constraint means that, while some small-brained species are able to produce large and/or frequent clutches, others can adopt a strategy of relatively slow reproductive output (Fig. 3b; lower quantile line: $\tau = 0.1$, $\beta \pm SE = -6.50 \pm 1.64$, $p \ll 0.001$). This small-brain-slow-output strategy is common in stable tropical regions but becomes progressively underrepresented, up to the point of exclusion, in high-latitude environments where conditions vary and winters are harsh (Fig. 4d–f).

**Environmental buffering via body size.** Morphological and physiological mechanisms that enhance an individual's ability to tolerate a wider range of temperatures can also be beneficial in thermally variable habitats[32]. Larger bodies, for example, are generally associated with broader thermal tolerances and a greater resistance to cold temperatures[33]. Additionally, bigger individuals may be better equipped to withstand long periods of fasting as those that are typically experienced during winter, because they exhibit lower per-mass energy expenditures[34] and a capacity to store more energy-rich fats[35]. In our global sample, we found that species of larger relative brain size show a slight tendency toward intermediate body sizes, while smaller-brained species show wider variation in body size (Fig. 3c; $n = 1304$; upper quantile regression line: $\tau = 0.9$, $\beta \pm SE = -0.85 \pm 0.17$ $p \ll 0.001$; lower quantile: $\tau = 0.1$, $\beta \pm SE = 1.28 \pm 0.04$, $p \ll 0.001$). When focusing our attention on the most thermally variable habitats, we find that the number of species with most trait combinations is no different from null expectations. The single exception is small-bodied,

small-brained species, which are almost entirely absent (Fig. 4g–i), suggesting that small-brained birds rely on large bodies to deal with the extreme winter conditions in these environments. Such a requirement may come from the need to be more metabolically efficient or capable of withstanding longer periods of low food abundance. However, it is also possible that small-brained birds need to be big instead because their low-quality diets require a sufficiently large gut for the efficient digestion of fibrous plant materials[36].

## Discussion

Our data suggest that resident birds in the most thermally variable and unpredictable habitats on Earth exhibit two alternative strategies for coping with environmental fluctuations. The most common strategy is consistent with the main premise of the cognitive buffer hypothesis, which is that enhanced capacity for behavioral flexibility facilitates coping with variable conditions[8,24,37–42]. Owing to the high demands of developing and maintaining large brains[20], the species that have adopted such strategy are constrained to high-quality diets and relatively low reproductive outputs[30]. Instead, the second and less common strategy emphasizes the ability to withstand or recover from environmental extremes by developing a large and expensive gut that can digest readily available fibrous plants, by producing a large number of offspring, and by having a large body size. This lifestyle appears to be largely incompatible with the strong metabolic demands of having a large brain[22], highlighting the important role that energetic constraints can play in mediating adaptation to extreme environments.

The observation that brain size bimodality is observed among resident but not migratory species in thermally variable environments suggests that extreme winter conditions play a primary role in driving the occurrence of these alternative strategies. Nevertheless, we note that, while species on the low end of the brain size spectrum could conceivably cope well with constant exposure to cold temperatures and scarce food, many Arctic and North Temperate birds with large brains alleviate winter food scarcity by accessing high-quality resources produced at other times of the year (e.g., through food caching[43,44], social foraging[45–47], or switching between foraging methods[48,49]). We therefore posit that, although extreme winter conditions are likely to be the main limiting factors in these systems, it is the frequent oscillation between harsh and mild conditions that favors bimodality in avian brain sizes at high-latitude environments. Specifically, environmental oscillations create environmental challenges that can be solved through either enhanced endurance and reproduction or through planning, memory, behavioral flexibility, and innovation. We also note that the complete absence of intermediate brain sizes among resident birds in thermally variable regions suggests that a fairly high level of investment in brain tissue is required to cope effectively with

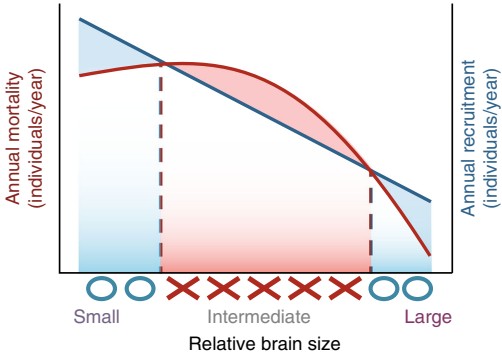

**Fig. 5** Potential relationships between relative brain size, annual mortality, and annual recruitment in high-latitude environments. The expected decrease in population recruitment as a function of brain size (blue line) is depicted as a linear function, as indicated by the patterns in Fig. 3b. In contrast, the effect of relative brain size on mortality rates (red line) is depicted as a quadratic function because intermediate brain sizes are likely to be inadequate for dealing with high environmental variability and can reduce the proportion of resources available for the deployment and use of other non-cognitive solutions to environmental variation. The example illustrated here demonstrates a scenario where populations of species with intermediate brain size would be excluded from thermally variable environments

winter challenges, suggesting that inferior investments are only sustainable if they are low enough to allow the development and maintenance of other costly coping mechanisms (Fig. 5).

Another interesting aspect of our findings is that the detected trade-offs between brain size, diet, and reproductive output are reflected in upper-bound constraints rather than in strictly optimal relationships[50]. Specifically, while big-brained birds are constrained to high-quality diets and low reproductive outputs, small-brained ones basically exhibit all kinds of diets and reproductive rates (Fig. 3a, b). We believe this observation may provide important clues on the possible origin of a small-brain strategy in high-latitude habitats. Specifically, previous work has shown that large-brained avian clades evolved in relatively stable environments and subsequently colonized variable ones[11]. Just like a bigger brain appears to have allowed corvids, owls, woodpeckers, and other clades to colonize highly variable environments through exaptation[51], the suite of traits we currently observe in small-brained birds in thermally variable regions is likely to have existed earlier in tropical areas and may have therefore enabled either the de novo colonization of temperate habitats or the persistence of these clades in formerly warm, stable regions that became cold and unpredictable as a product of climate change[52]. The tropical origin of Galliformes and their subsequent success in temperate and polar environments support such a habitat filtering scenario[53].

In conclusion, this study highlights the complex and often underappreciated role of climate in shaping the global distribution of brain size in birds. Our analyses strongly support the notion that habitat filtering related to brain size is relatively weak in the tropics but strong at high latitudes where the challenges of coping with harsh winter months drive the exclusion of resident species with intermediate brain sizes[54–56]. These results remind us that, while big brains enable coping with the occasional challenges of harsh winters in highly seasonal environments, small ones can be equally effective in addressing those challenges when paired with other natural history traits, such as the capacity to endure food scarcity and/or the ability to produce large numbers of offspring. Ultimately, the pattern that we report here indicates that functional diversity is reduced in temperate and

arctic habitats, given the narrower range of trait combinations that can effectively counter the challenges that are often encountered during extreme winters.

## Methods

**Quantifying relative brain size.** Absolute brain size generally increases with body size in vertebrates[13], but experimental and comparative studies have shown that, in birds, accounting for body size when measuring brain size can provide a better indication of certain aspects of cognition such as problem solving and behavioral innovation[37,57–60]. Accordingly, we accounted for the sublinear allometric relationship between brain and body size in our analyses by explicitly working with relative brain size, measured as the residuals from a phylogenetic generalized least squares (PGLS) log-log regression model of brain size on body size (Supplementary Fig. 3). This allometric model included data for all the previously published avian brain sizes, $n = 2062$ species[11], and accounted for phylogenetic uncertainty by being independently run on 1000 randomly selected tree topologies with the Hackett backbone (mean slope $\pm$ s.d. = 0.594 $\pm$ 0.001; mean intercept $\pm$ s.d. = $-1.08 \pm 0.02$; mean $\lambda \pm$ s.d. = 0.87 $\pm$ 0.01; www.birdtree.org; downloaded July 2016)[61]. We note that relative brain size values used in subsequent analyses are the median residuals for each species across these 1000 models. Acknowledging that debate on the interpretation of results derived from phylogenetic comparative methods remains unresolved[62,63], we note that brain size residuals computed from PGLS are highly correlated to those computed through ordinary least squares (OLS) regression ($r = 0.99$; OLS regression slope $\pm$ SE = 0.563 $\pm$ 0.004; intercept $\pm$ SE = $-0.852 \pm 0.01$) and show in our Supplementary Fig. 4 and Supplementary Tables 2–6 that our results remain unchanged when the latter method is used. The only exception was in the case of the upper constraint imposed on percentage of vegetative diet by relative brain size, which was in the same direction but no longer met criteria for significance (upper quantile regression line: $\tau = 0.9$, $\beta \pm$ SE = $-20.18 \pm 10.78$, $p = 0.06$; Supplementary Fig. 4a). While the brain comprises a number of different regions that vary in function, previous work in birds has shown that changes in size tend to be concerted among regions and that interspecific differences in relative brain size are typically associated with volumetric variation in regions associated with executive function[38,64,65]. Furthermore, current evidence indicates that larger brain sizes are specifically associated with increased neuron numbers, suggesting that large relative brain size corresponds to higher neuron numbers for a given body size[66]. We therefore follow previous works that suggest relative size of the whole brain provides a useful, albeit imperfect (see ref. [67]), indicator of cognitive ability for broad-scale comparative analyses[9,10,39,57,60,68].

**Quantifying environmental variability.** Previous work has shown that environmental conditions become increasingly variable when moving from the tropics toward higher-latitude environments[9,11]. We captured this gradient with the first component (PC1) recovered from geographic principal component analysis (PCA), computed from a 100+ year time series of local means, seasonalities, and predictabilities of precipitation and temperature[16]. The time series used in this analysis had a global coverage (excluding Antarctica) and was downloaded at a spatial resolution of 0.5° × 0.5° and subsequently transformed to Wagner IV equal area projection. Seasonality was measured as the within-year variance of temperature and the coefficient of variation across monthly values for precipitation. Predictability was measured as Colwell's $P$, an index that captures among-year variation in the onset, intensity, and duration of periodic phenomena[69]. All environmental variables involved were transformed when required and subsequently centered and scaled prior to PCA[70].

**The global distribution of brain size.** We assessed the association between environment and relative brain size by testing whether frequency distributions of relative brain size differed from null expectations in environmental regions spanning the global gradient of environmental conditions captured by PC1. Six environmental regions were delimited by dividing the range of PC1 values into equal bins. Frequency distributions of relative brain size were subsequently characterized by considering all species whose breeding range overlapped with each of these regions[71]. We note that environmental regions differ in geographic area because the coverage of different types of environmental conditions differs across the world. For each environmental region, we compared kernel density estimates of relative brain size derived from empirically observed species[72] to the expected density estimates derived from 10,000 random samples of species. Specifically, we counted the observed number of species in the region of interest ($n$) and estimated kernel densities of relative brain size for 10,000 samples of $n$ species randomly selected from the global species pool in our sample[54]. Then, at 15 evenly spaced points across the global range of relative brain sizes, we computed $z$-scores to test whether the empirically derived density estimate differed significantly from the distribution of expected density estimates obtained through randomization. Within each environmental bin, the 15 $p$ values produced through this procedure were corrected for multiple comparisons using Holm's method[73]. Because migrant and resident birds vary in their exposure to the range of conditions encountered within their breeding habitats, we analyzed their brain size distributions separately. Pelagic species (orders Sphenisciformes, Suliformes, Procellariiformes, and

Phaethontiformes and families Pelecanidae, Laridae, Stercorariidae, and Alcidae) were excluded from analyses because land surface temperature and precipitation data are unlikely to reflect the conditions that species that spend much of their time at sea typically encounter.

**Quantifying environmental constraints on traits**. We used quantile regression to quantify whether relative brain size can impose upper- or lower-bound constraints on reproductive output, percentage of vegetative material within a diet, and body size in our global sample of resident species[74]. Upper bounds were estimated by setting $\tau$ at 0.9, whereas lower bounds were estimated by setting it at 0.1. Slopes differing significantly from zero at either of these levels are considered here as indicative of a constraint. Body size and reproductive output were obtained from ref.[75], with the latter calculated as the number of eggs per clutch multiplied by the number of clutches per year. Percentage of vegetative plant material in diet was obtained from ref.[76] and was computed from plant material consumption other than fruit, seeds, or nectar.

Finally, to test whether environmental conditions constrained the viability of certain trait combinations, species were categorized as having either small or large relative brain size, low or high percentage of vegetative material in their diet, low or high reproductive output, and small or large body size based on the corresponding global means of these traits. For all species whose breeding distribution overlapped with a given environmental region, we counted the number of cases with each trait combination and compared these numbers to randomized (null) expectations. In this case, we used a sliding window that spanned one sixth of the range of environmental PC1 and captured 20 partially overlapping environmental regions of the globe from the Tropics to the Arctic. For each environmental region, null expectations were derived by drawing 10,000 samples of $n$ species from the global species pool, where $n$ was the number of species observed in the region of interest. We then computed $z$-scores to test whether the observed number of species with each trait combination differed from the expected distribution of counts derived from randomizations.

**Reporting summary**. Further information on research design is available in the Nature Research Reporting Summary linked to this article.

## Data availability

All data generated or analyzed during this study are included in this published article (and its supplementary information files). The source data underlying Figs. 1, 2, 3, and 4 and Supplementary Figs. 1, 3, and 4 are provided as a Source Data file.

## Code availability

The R code required to rerun analyses is provided as Supplementary Data 1.

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

## Acknowledgements

We thank Andrew Iwaniuk and the National Evolutionary Synthesis Center for guidance and for compiling the original brain size data set. Additionally, we thank Bruno Vilela and Zhijie Zhang for advice on data presentation and Joseph R Burger for helpful discussion. C.A.B. was funded by NSF, under award number DEB 1841470.

## Author contributions

Both authors designed the study. T.S.F. performed analyses and prepared the initial draft of the manuscript. Both authors contributed significantly to subsequent revisions.

## Additional information

**Competing interests:** The authors declare no competing interests.

