## [Peer Review File · Nature Communications]

Reviewers' Comments:

Reviewer #1:

Remarks to the Author:

This manuscript is an excellent and much needed study on the correlates of brain size evolution in birds. This study is highly original, carefully done and very important for a broad readership.

It is well written and of appropriate length, the structure is clear and the methods are described in detail. The figures are a bit complex first, but very nicely done and helpful for understanding the patterns. Especially helpful are the R scripts and the complete data file.

The statistical approach, including the PGLS analysis, is correct and state-of-the-art. Actually, the chosen approach does not fully investigate on which level the variation in relative brain size is located, deep down at the splits between families (where much of the variation in bird brain size is located), or in the terminal branches of the phylogenetic tree, as species-specific adaptations (where error variation may obscure the patterns). However, I think the quantile regressions applied here are an elegant and useful way to deal with the problem of too much or too little correction for phylogenetic relatedness.

While some people criticize the approach of phylogenetic comparative studies altogether, I think the study presented here is an excellent example of the power of this approach. By carefully looking into patterns, we can recognize potential constraints and pathways, and go well beyond speculations. Of course, evolutionary hypotheses are difficult to test, and the comparative approach is just one possibility. However, here it is applied in a very professional and thorough manner. Looking at whole brain size is perfectly appropriate for a study on this level, as interindividual variation is considerable and a large sample size is therefore crucial.

In sum, I highly recommend publication of this paper in its present state.

Perhaps one addition should be made: van Woerden et al. find evidence for an energetic constraint of high experienced seasonality on relative brain size in the (non-migratory) primates (e.g. van Woerden et al. 2014, *Am. J. of Phys. Anthropol.*). Although concerning primates, it would be relevant to cite these findings to strengthen the broad importance of the results presented here.

Reviewer #2:

Remarks to the Author:

This is a well written and highly interesting manuscript on how relative brain size varies across different habitats in a large sample of different species of birds. The study identifies a very interesting and completely novel pattern in that intermediate levels of relative brain size are scarce in harder (here colder) climates. Instead, the data points towards two main strategies, either species in harsh environments have a large brain in relation to body size (presumably a combination leading to high cognitive abilities allowing for instance foraging under difficult conditions) or they have small relative brain size in a larger body (presumably allowing better protection against cold temperatures while having higher reproductive output). I think the study is very well orchestrated and many of the potential confounders have been adequately dealt with (phylogenetic relationships, phylogenetic uncertainty, body size effects (but see additional comment on this below), quantification of environmental variability, effects of other variables, etc). The study also forms a very solid comparative correlative empirical background that will elicit many possible follow up projects on for instance i) within species variation in relative brain size across variable environments, ii) variation in resource allocation between neural aspects and other traits in different environments and iii) the overall concept of focusing not only on increases in relative brain size but also on potentially adaptive decreases in relative brain size.

One potential addition to the text stems from the fact that body size has a strong association with both absolute brain size (discussed by the authors), and relative brain size (not discussed by the authors). The well-known negative allometry of brain and body size usually means that smaller bodied animals tend to have large relative brain size, while larger bodied animals tend to have smaller relative body size. This aspect could be brought up in the paragraph going through the quantification of relative brain size. Luckily the authors have body size as a separate factor in the analyses where they compare all combinations of relative brain size and body size, so this potential confounder is not driving these results. But I would still encourage the authors to bring up the allometry issue briefly because it is a frequently discussed issue in the field.

There is also still some debate over what the functional consequences of a larger brain really are. Several recent studies have demonstrated experimentally that larger brains are associated with higher cognitive abilities (the authors cite some of these). Increased number of neurons in larger brains is often proposed as the mechanistic background to why larger brains mean higher cognitive abilities. Because of this I think the authors should also add a sentence or two on that larger relative brain size often means higher number of neurons for a given body size. There is certainly citable data supporting this pattern.

Overall, I have no other comments on the manuscript. I think it is an excellent story and I congratulate the authors on a very interesting and important study.

Reviewer #3:

Remarks to the Author:

The authors of this paper address a question that has interested many previous authors (resulting in many dozens of papers): why brain size varies. Here they suggest that trade-offs between selection on body and brain size is an important consideration and that this is driven by the climate of high latitudes where there is 'habitat filtering related to brain size'. The take home message is: "big brains enable coping with the occasional challenges of harsh winters in highly seasonal environments, small ones can be equally effective in addressing those challenges when paired with other natural history traits such as the capacity to endure food scarcity and/or the ability to produce large numbers of offspring."

The question is, how novel and substantial is this new addition to the already-considerable wealth of literature on this issue? Until this manuscript, the outstanding questions had been: 1) how to reconcile the multiple explanations for variation in brain size (in various vertebrate taxa); 2) what cognitive advantage does brain size confer on its owner? This latter question has only been addressed in the comparative literature in the form of assumptions (e.g. innovation is an indicator of 'intelligence') or is assumed (as in this manuscript) to be an indicator of "planning, innovation, and behavioral flexibility", although explicit cognitive assessment of these and proxy for them is incorporated here: it is simply assumed that larger brains provide such benefits. Here the authors appear to promise to examine this question (L 46-49) but don't actually say anything about cognition (other than to make the same assumptions made by previous authors). Question 1 is also not addressed here either. Rather the authors suggest that the problem that has been missed is a focus on relatively small brains (which have always been included in the previous analyses, it is just that no one has thought to pay much attention to the data below the regression line, according to these authors). If we did, then, we would take a significant step forward in understanding brain size evolution. This may well be the case. Indeed, although the authors don't say this directly, their findings suggest that being smart (the typical assumption of the benefit of a larger brain) is not the only way to be successful in harsh

northern climates. This would

The addition the authors would like to make is that at high altitudes relative brain size is either big or small. This bimodality is what they then go on to discuss for the remainder of their paper. One of these suggestions that these birds are making a trade-off by allocating 'greater proportion of their limited budget'. It seems moderately relevant to have this idea 'unpacked': what is this budget? Is it energy?

It would be useful for the authors to supply a figure with body size and brain size plotted, not the figure provided in the Supplementary Materials that has body against relative brain size (why would this be plotted given that relative brain size already has an effect of body size removed?). Given that the authors consider birds might trade these two variables off against one another, it seems relevant to see how these variables map onto one another: there are multiple ways to get a relatively small brain and one needs to get an idea whether selection might have been on body size or on brain size. This difficulty has been seen before in the domestication literature where species have relatively smaller brains - the selection might have come from increasing body size, decreasing brain size or both. It is still not clear which of these is the appropriate explanation in the case of domestication so these authors need to make it very clear what they consider selection is acting on: body size or brain size. Indeed, lines 189-197 suggest that the authors can't tell.

Also it seems important the brain size and energetic demands are related and yet although on L206 the authors cite the Isler and Schaik (2006) paper in that paper the authors could find no relationship between brain size and BMR in birds. How then, does this citation support this comment: "Due to the high demands of developing and maintaining large brains..."?

REVIEWERS' COMMENTS: _____

Reviewer #1 (Remarks to the Author):

This manuscript is an excellent and much needed study on the correlates of brain size evolution in birds. This study is highly original, carefully done and very important for a broad readership.

It is well written and of appropriate length, the structure is clear and the methods are described in detail. The figures are a bit complex first, but very nicely done and helpful for understanding the patterns. Especially helpful are the R scripts and the complete data file.

The statistical approach, including the PGLS analysis, is correct and state-of-the-art. Actually, the chosen approach does not fully investigate on which level the variation in relative brain size is located, deep down at the splits between families (where much of the variation in bird brain size is located), or in the terminal branches of the phylogenetic tree, as species-specific adaptations (where error variation may obscure the patterns). However, I think the quantile regressions applied here are an elegant and useful way to deal with the problem of too much or too little correction for phylogenetic relatedness.

While some people criticize the approach of phylogenetic comparative studies altogether, I think the study presented here is an excellent example of the power of this approach. By carefully looking into patterns, we can recognize potential constraints and pathways, and go well beyond speculations. Of course, evolutionary hypotheses are difficult to test, and the comparative approach is just one possibility. However, here it is applied in a very professional and thorough manner. Looking at whole brain size is perfectly appropriate for a study on this level, as interindividual variation is considerable and a large sample size is therefore crucial.

In sum, I highly recommend publication of this paper in its present state. Perhaps one addition should be made: van Woerden et al. find evidence for an energetic constraint of high experienced seasonality on relative brain size in the (non-migratory) primates (e.g. van Woerden et al. 2014, Am. J. of Phys. Anthropol). Although concerning primates, it would be relevant to cite these findings to strengthen the broad importance of the results presented here.

We thank the reviewer for their support of our work and for pointing out the van Woerden paper. It is certainly of relevance to our work and we have cited it in lines 166 and 255.

Reviewer #2 (Remarks to the Author):

This is a well written and highly interesting manuscript on how relative brain size varies across different habitats in a large sample of different species of birds. The study identifies a very interesting and completely novel pattern in that intermediate levels of relative brain size are scarce in harder (here colder) climates. Instead, the data points towards two main strategies, either species in harsh environments have a large brain in relation to body size (presumably a combination leading to high cognitive abilities allowing for instance foraging under difficult conditions) or they have small relative brain size in a larger body (presumably allowing better protection against cold temperatures while having higher reproductive output). I think the study is very well orchestrated and many of the potential confounders have been adequately dealt with (phylogenetic relationships, phylogenetic uncertainty, body size effects (but see additional comment on this below), quantification of environmental variability, effects of other variables, etc). The study also forms a very solid comparative correlative empirical background that will elicit many possible follow up projects on for instance i) within species variation in relative brain size across variable environments, ii) variation in resource allocation between neural aspects and other traits in different environments and iii) the overall concept of focusing not only on increases in relative brain size but also on potentially adaptive decreases in relative brain size.

One potential addition to the text stems from the fact that body size has a strong association with both absolute brain size (discussed by the authors), and relative brain size (not discussed by the authors). The well-known negative allometry of brain and body size usually means that smaller bodied animals tend to have large relative brain size, while larger bodied animals tend to have smaller relative body size. This aspect could be brought up in the paragraph going through the quantification of relative brain size. Luckily the authors have body size as a separate factor in the analyses where they compare all combinations of relative brain size and body size, so this potential confounder is not driving these results. But I would still encourage the authors to bring up the allometry issue briefly because it is a frequently discussed issue in the field.

Because we calculate relative brain size as the residuals of the log-log relationship between brain and body size, this metric accounts for the sublinear (negative allometric) relationship described by the reviewer. We have now specifically clarified that the allometric relationship between brain and body size is sublinear in the section 'Quantifying relative brain size' (lines 338-346) and included supplementary fig. 3 to show this relationship.

There is also still some debate over what the functional consequences of a larger brain really are. Several recent studies have demonstrated experimentally that larger brains are associated with higher cognitive abilities (the authors cite some of these). Increased number of neurons in larger brains is often proposed as the mechanistic background to why larger brains mean higher cognitive abilities. Because of this I think the authors should also add a sentence or two on that larger

relative brain size often means higher number of neurons for a given body size. There is certainly citable data supporting this pattern.

We added a sentence on this supporting evidence in the 'Quantifying relative brain size' section and cited relevant research (lines 362-405).

Overall, I have no other comments on the manuscript. I think it is an excellent story and I congratulate the authors on a very interesting and important study.

We thank the reviewer for their suggestions and their support.

Reviewer #3 (Remarks to the Author):

The authors of this paper address a question that has interested many previous authors (resulting in many dozens of papers): why brain size varies. Here they suggest that trade-offs between selection on body and brain size is an important consideration and that this is driven by the climate of high latitudes where there is 'habitat filtering related to brain size'. The take home message is: "big brains enable coping with the occasional challenges of harsh winters in highly seasonal environments, small ones can be equally effective in addressing those challenges when paired with other natural history traits such as the capacity to endure food scarcity and/or the ability to produce large numbers of offspring."

The question is, how novel and substantial is this new addition to the already-considerable wealth of literature on this issue? Until this manuscript, the outstanding questions had been: 1) how to reconcile the multiple explanations for variation in brain size (in various vertebrate taxa); 2) what cognitive advantage does brain size confer on its owner? This latter question has only been addressed in the comparative literature in the form of assumptions (e.g. innovation is an indicator of 'intelligence') or is assumed (as in this manuscript) to be an indicator of "planning, innovation, and behavioral flexibility", although explicit cognitive assessment of these and proxy for them is incorporated here: it is simply assumed that larger brains provide such benefits. Here the authors appear to promise to examine this question (L 46-49) but don't actually say anything about cognition (other than to make the same assumptions made by previous authors).

We fully agree with the principle that a better understanding of the cognitive advantages that are awarded by increased brain size is an important research goal and have been striving to accomplish that in other projects currently underway. However, for this particular paper, we rely on published work (see lines 405-408) to establish a reasonable connection between cognition and relative brain size and use it as a springboard to answer a very different set of questions: how does the distribution of relative brain size in birds vary geographically and what possible mechanisms explain what we see? We strongly believe that this kind of analyses is highly complementary and quite useful to such a research program. For example, our analyses highlight that our field's current focus on explaining brain size enhancements may be preventing us from seeing the larger picture and from

understanding the true set of constraints and issues that ultimately influence the outcome of brain evolution in a clade.

Question 1 is also

not addressed here either. Rather the authors suggest that the problem that has been missed is a focus on relatively small brains (which have always been included in the previous analyses, it is just that no one has thought to pay much attention to the data below the regression line, according to these authors). If we did, then, we would take a significant step forward in understanding brain size evolution. This may well be the case. Indeed, although the authors don't say this directly, their findings suggest that being smart (the typical assumption of the benefit of a larger brain) is not the only way to be successful in harsh northern climates. This would We fully agree that our results indicate that there are different ways to accomplish persistence in highly variable environments and have made our best effort to say that in the title, introduction, results and discussion sections of our paper. However, we strongly disagree with the reviewer's implication that this observation is not new or interesting. These ideas have simply not been mentioned (or let alone been tested!) in prior literature, and we hope they will help set future agenda for investigating the evolution of cognition in a more holistic fashion.

The addition the authors would like to make is that at high altitudes relative brain size is either big or small. This bimodality is what they then go on to discuss for the remainder of their paper. One of these suggestions that these birds are making a trade-off by allocating 'greater proportion of their limited budget'. It seems moderately relevant to have this idea 'unpacked': what is this budget? Is it energy? The text discusses both the energetic and developmental costs of large brain size and cites relevant literature that demonstrates these costs (e.g. lines 141-142 or 204-211). We now specify 'energetically costly' when discussing the digestion of fibrous food items (line 170) as the text was previously unclear here. By the way, while our findings could potentially be extrapolated to altitude, we note that the text explicitly states that geographic patterns we talk about are primarily linked with latitude.

It would be useful for the authors to supply a figure with body size and brain size plotted, not the figure provided in the Supplementary Materials that has body against relative brain size (why would this be plotted given that relative brain size already has an effect of body size removed?).

Supplementary figure 3 has been added showing the allometry between brain mass and body mass. The supplementary figure that the reviewer is referring to (which also has a version using PGLS residuals in the main text – Fig. 3c) is included for consistency with the other two traits (diet and reproductive output) and because we later show how the filling of the trait space depicted in this figure is constrained by environmental conditions (through the analyses depicted in figure 4g-i).

Given that the authors consider birds might trade these two variables off against one another, it seems relevant to see how these variables map onto one another:

there are multiple ways to get a relatively small brain and one needs to get an idea whether selection might have been on body size or on brain size. This difficulty has been seen before in the domestication literature where species have relatively smaller brains - the selection might have come from increasing body size, decreasing brain size or both. It is still not clear which of these is the appropriate explanation in the case of domestication so these authors need to make it very clear what they consider selection is acting on:

body size or brain size. Indeed, lines 189-197 suggest that the authors can't tell.

Because our data shows that species of any body size may possess either large or small relative brain sizes, we are not suggesting that species trade-off between brain and body size. Instead we are making the point that small-bodied species with small relative brain size do not inhabit high latitude environments year round (see figure 4g-i). It is true that our analyses do not allow us to determine whether selection in these species has acted directly on brain or body size or both (and we even suggest in our discussion at lines 295-304 that the evolution large or small relative brain sizes may not be initially driven by climatic variability). However, this is not a problem for our current study as we are considering the ecological and biogeographic consequences of different relative brain sizes rather than their evolutionary origins.

Also it seems important the brain size and energetic demands are related and yet although on L206 the authors cite the Isler and Schaik (2006) paper in that paper the authors could find no relationship between brain size and BMR in birds. How then, does this citation support this comment: "Due to the high demands of developing and maintaining large brains...?"

We have added a citation for a study that more directly demonstrates the energetic requirements of brain tissue at line 142. However, we also point out that our study is directly interested in energetic trade-offs, which are consistent with evidence from Isler and Schaik (2006) that the energy budget (indicated by BMR) does not increase with brain size. With a consistent budget, resources allocated to increased brain tissue must necessarily decrease investment elsewhere (e.g. Gut tissue).